# Adversarial Interaction Attacks: Fooling AI to Misinterpret Human Intentions

Nodens Koren [1]  Xingjun Ma [2]  Qiuhong Ke [1]  Yisen Wang [3]  James Bailey [1]

## Abstract

Understanding the actions of both humans and artificial intelligence (AI) agents is important before modern AI systems can be fully integrated into our daily life. In this paper, we show that, despite their current huge success, deep learning based AI systems can be easily fooled by subtle adversarial noise to misinterpret the intention of an action in interaction scenarios. Based on a case study of skeleton-based human interactions, we propose a novel adversarial attack on interactions, and demonstrate how DNN-based interaction models can be tricked to predict the participants' reactions in unexpected ways. Our study highlights potential risks in the interaction loop with AI and humans, which need to be carefully addressed when deploying AI systems in safety-critical applications.

## 1. Introduction

State-of-the-art action recognition and prediction models are deep neural networks (DNNs), due to their capability of modeling complex problems (Si et al., 2019; Li et al., 2019a;b) in an accurate way. Nonetheless, it has also been shown that these models are prone to adversarial examples (or attacks). DNNs can behave erratically when processing inputs with carefully crafted perturbations, even though such perturbations are imperceptible to humans (Biggio et al., 2013; Szegedy et al., 2013; Goodfellow et al., 2014).

In this work, we investigate the adversarial vulnerability of DNN reaction prediction (i.e., regression) models in skeleton-based interactions. Skeleton signals are among one of the most commonly used representations for human or robot motion (Zhang et al., 2016; Wang et al., 2018). While

[1]School of Computing and Information Systems, University of Melbourne, Parkville, VIC, Australia [2]School of Information Technology, Deakin University, Geelong, VIC, Australia [3]School of EECS, Peking University, Beijing, China. Correspondence to: Xingjun Ma <daniel.ma@deakin.edu.au>.

*Accepted by the ICML 2021 workshop on A Blessing in Disguise: The Prospects and Perils of Adversarial Machine Learning.* Copyright 2021 by the author(s).

adversarial attacks have been extensively studied on images (Goodfellow et al., 2014; Su et al., 2019; Brown et al., 2017; Duan et al., 2020), very few works have been proposed for skeletons (Liu et al., 2019; Wang et al., 2019; Zheng et al., 2020). In comparison to the image space, which is continuous and where pixels can be perturbed freely without raising obvious attack suspicions, the skeleton space is sparse and discrete. It has a temporal nature that needs to be taken into account. Consequently, attacking skeleton-based models requires many more constraints than the image space.

Existing work on attacking skeleton-based models have only considered the single-person scenario, and have all focused towards recognition (i.e., classification) models (Liu et al., 2019; Wang et al., 2019; Zheng et al., 2020). However, interaction scenarios involving two or more characters are essential to the interaction between humans and AI. They should not be overlooked if our ultimate goal is to build AI agents that can fit into our daily life.

To close this gap, we propose an Adversarial Interaction Attack (AIA) to test the vulnerability of regression DNNs in skeleton-based interactions involving two characters. Being able to recognize a person's action accurately is important, but it is equally important to be able to go a step further and *respond* to the action in an appropriate way. In light of this, the usage of regression models is necessary. We hence modified the output layers of two state-of-art models on action recognition to return reactor sequences instead of class labels, and we trained them on skeleton-based interaction data. We examine the performance of AIA under both white-box and black-box settings. We show that our AIA attack can easily fool the two regression models to misinterpret the actor's intentions and predict unexpected reactions. Such reactions have detrimental effects on either the actor or the reactor. Overall, our work reveals potential threats of subtle adversarial attacks on interactions involving AI.

In summary, our contributions are:

- We propose an adversarial attack approach - Adversarial Interaction Attack (AIA), that is domain-independent, and has the potential to work for general sequential regression models.

- We propose an evaluation metric that can be applied to evaluate the performance of sequential regression

attacks. Such a metric is currently missing from the literature.

- We empirically show that our AIA attack can generate targeted adversarial action sequences with small perturbations, which fool DNN regression models into making incorrect (possibly dangerous) predictions.

Note that the goal of our work is to design a new type of attack and evaluation metric that is capable of handling any type of regression-based problems in general. We thus leave the compatibility between our work and previously proposed anthropomorphic constraints (Liu et al., 2019; Wang et al., 2019; Zheng et al., 2020) as a future area of interest.

## 2. Proposed Adversarial Interaction Attack

In this section, we first provide a mathematical formulation of the targeted adversarial sequence attack problem. We then introduce the loss functions used by our AIA attack.

**Overview.** Intuitively, the goal of our AIA attack is to deceive the *reactor* AI agent into thinking that the *actor* is doing a different specific action by making minor changes to the positions of the *actor's* joints or the angles between joints. The reactor agent will consequently respond by performing the reaction that is targeted by the attack.

### 2.1. Formal Problem Definition

A skeleton sequence with $T$ frames can be represented mathematically as the vector $\mathbf{X} = (\mathbf{x}_1, \mathbf{x}_2, ..., \mathbf{x}_T)$ where $\mathbf{x}_i$ is a skeleton representation of the $i^{th}$ frame, which is a vector consists of 3D-coordinates of the human skeleton joints. More specifically, $\mathbf{x}_i \in R^{N \times 3}$, where N denotes the number of the joints. In our approach, we flattened $\mathbf{x}_i$ into $R^{3N}$.

First, we define the formal notion of interaction. Suppose the two characters in a two-person interaction scenario are *actor A* and *reactor B*. The task of an interaction prediction model $f$ is to predict an appropriate reaction (i.e., skeleton) $\mathbf{y}_t$ at each time step $t$ for reactor B based on the observed skeleton sequence of actor A $(\mathbf{x}_1, \cdots, \mathbf{x}_t)$. This can be written mathematically as:

$$f(\mathbf{x}_1, \cdots, \mathbf{x}_{t-1}, \mathbf{x}_t) = \mathbf{y}_t.$$

Given an input skeleton sequence $\mathbf{X} = (\mathbf{x}_1, \mathbf{x}_2, ..., \mathbf{x}_T)$, an adversarial target skeleton sequence $\mathbf{Y}' = (\mathbf{y}'_1, \mathbf{y}'_2, ..., \mathbf{y}'_T)$, and a prediction model $f : R^{T \times 3N} \to R^{T \times 3N}$, the goal of our AIA attack is to find an adversarial input sequence $\mathbf{X}' = (\mathbf{x}'_1, \cdots, \mathbf{x}'_T)$ by solving the following optimization problem:

$$\min_{\mathbf{X}'} \sum_{t \in T} \|\mathbf{x}'_t - \mathbf{x}_t\|_{\infty} \, s.t. \sum_{t \in T} \|f(\mathbf{x}'_1, \cdots, \mathbf{x}'_t) - \mathbf{y}'_t\|_2 < \kappa, \tag{1}$$

where, $\| \cdot \|_p$ is the $L_p$ norm, and $\kappa \geq 0$ is a *tolerance factor*, which serves as a cutoff that distinguishes whether the output sequence is recognizable as the target reaction. This gives us more flexibility when crafting the adversarial input sequence $\mathbf{X}'$ because the acceptable target sequence is non-singular; the output sequence does not need to be exactly the same as the target sequence to resemble a particular action. We empirically determine this factor based on informal user survey in Appendix B.1. Intuitively, the above objective is to find a sequence $\mathbf{X}'$ with minimum perturbation from $\mathbf{X}$, such that the distance between the output and the target is less than $\kappa/T$ on average for each time step.

### 2.2. Adversarial Loss Function

Our goal is to develop a mechanism that crafts an adversarial input sequence which solves the above optimization problem given any target output sequence, while also maintaining the naturalness of the adversarial input sequence. In order to achieve this goal, we propose the following adversarial loss function:

$$\mathcal{L}_{adv} = \mathcal{L}_{spatial} + \lambda \mathcal{L}_{temporal}, \tag{2}$$

where the $\mathcal{L}_{spatial}$ loss term minimizes the spatial distance between the output sequence and the target sequence, and the $\mathcal{L}_{temporal}$ loss term maximizes the coherence of the perturbed input sequence so as to maintain the naturalness of the adversarial input sequence.

**Spatial Loss.** The spatial loss term aims to generate adversarial output sequences that are visually similar to the target reaction sequences; that is, its objective is to minimize the spatial distance between the output joint locations and the *neighbourhood* of the target joints for every time step. Following the formulation of the relaxed optimization problem in Equation 1, we use the $L_2$ norm to measure the distance between two sets of joint locations:

$$\mathcal{L}_{spatial} = \sum_{t \in T} \inf\{\|f(\mathbf{x}'_1, \cdots, \mathbf{x}'_t) - \mathbf{p}_t\|_2 \mid \mathbf{p}_t \in S_t\} \tag{3}$$

with $S_t$ being an $(N\text{-}1)$-sphere defined by:

$$S_t(\mathbf{y}'_t, \eta) = \{\mathbf{p}_t \in R^{3N} \mid \|\mathbf{p}_t - \mathbf{y}'_t\|_2 = \eta\}. \tag{4}$$

Here, $\eta = \kappa/T$ is the mean of the enabling tolerance factor $\kappa$ in equation Equation 1 over time $T$.

**Temporal Loss.** The temporal loss term is to guarantee the naturalness of the generated adversarial input sequence.

Specifically, the movement of each joint should be continuous in time, and motions with abrupt huge change or teleportation should be penalised. The $\mathcal{L}_{temporal}$ term achieves this goal by maximizing the coherence of each element in the perturbed input sequence with respect to its neighboring elements in the temporal dimension. This gives:

$$\mathcal{L}_{temporal} = \sum_{t \in T}(\|\mathbf{x}'_t - \mathbf{x}'_{t-1}\|_2 + \|\mathbf{x}'_t - \mathbf{x}'_{t+1}\|_2) \quad (5)$$

A scaling factor $0 \leq \lambda \leq 1$ is introduced in front of $\mathcal{L}_{temporal}$ to balance the two loss terms.

We use the first-order method Project Gradient Descent (PGD) (Madry et al., 2018) to minimize the combined adversarial loss iteratively as follows:

$$\mathbf{X}'_0 = \mathbf{X}\mathbf{X}'_{m+1} = \Pi_{\mathbf{X},\epsilon}\big(\mathbf{X}'_m - \alpha \cdot (\nabla_{\mathbf{X}'_m}\mathcal{L}_{adv}(\mathbf{X}'_m, \mathbf{Y}'))\big) \quad (6)$$

where, $\Pi_{\mathbf{X},\epsilon}(\cdot)$ is the projection operation that clips the perturbation back to $\epsilon$-distance away from $\mathbf{X}$ when it goes beyond, $\nabla_{\mathbf{X}'_m}\mathcal{L}_{adv}(\mathbf{X}'_m, \mathbf{Y}')$ is the gradient of the adversarial loss to the input sequence, $m$ is the current perturbation step for a total number of $M$ steps, $\alpha$ is the step size and $\epsilon$ is the maximum perturbation factor. The sequence $\mathbf{Y}'$ for a target reaction can be either customized or sampled from the original dataset.

## 3. Overview on a Case Study

In this section, we conduct a study on a selected set of attack objectives that can be easily associated with real scenarios and can serve as motivations behind our approach. We provide two extra case studies in Appendix A. Detailed experimental settings can be found in Section 4.

### 3.1. Case Study: 'punching' to 'handshaking'

In this case study we consider a case opposite to the previous one, where human exploiters are capable of attacking AI agents actively and derive benefit from being active attackers. In the future, it could become a common practice to utilize AI agents to complete dangerous tasks so as to lower the chance of human operators incurring injuries or fatalities. Security guard is one such job that might be taken over by an AI agent. Imagine a secret agency that hires AI security guards is invaded by intruders and is placed in a scenario where combat becomes necessary. The AI guard will fail in its role if the invaders know how to apply effective adversarial attacks towards it. This is the case in Figure 1 where the model was fooled to suggest 'handshaking' for the reactor (the green character) rather than 'punching'.

## 4. Performance Evaluation

In this section, we conduct an experiment to evaluate the effectiveness (white-box attack success rate) of our AIA at-

tack. We conduct an extra experiment on the transferability (black-box attack success rate) in Appendix C.

### 4.1. Experimental Settings

**Dataset.** We conduct our experiments on the SBU Kinect Interaction Dataset, which is composed of interactions of eight different categories, namely 'approaching', 'departing', 'kicking', 'punching', 'pushing', 'hugging, 'handshaking', and 'exchanging'. It contains 21 sets of data sampled from 7 participants using a Microsoft Kinect sensor, with approximately 300 interactions in total. Each character's information is encoded into 15 joints with the $x$, $y$, and depth dimensions. The values of $x$ and $y$ fall within $[0, 1]$, and depth in $[0, 7.8125]$.

We partitioned each interaction into two individual sequences corresponding to each character, respectively. One sequence will be used as the action (input), and another will be used as the reaction (output). Due to the lack of data in this dataset, we trained our response predictors from both characters' perspectives. With this belief, we used the skeleton sequences of both characters as input data independently. That is, for each interaction sequence $\mathbf{x} = \mathbf{x}_1 \frown \mathbf{x}_2$, we create two input/target pairs $(\mathbf{x}_1, \mathbf{x}_2)$ and $(\mathbf{x}_2, \mathbf{x}_1)$.

**Models and Training.** We adopted one convolutional model, TCN (Bai et al., 2018), and one recurrent model, DeepGRU (Maghoumi & LaViola Jr, 2019), and modified them such that the models predict sequences instead of categorical labels. Our TCN model has 10 hidden layers with 256 units in each layer, and our DeepGRU model follows Maghoumi & LaViola Jr (2019) exactly, with the output being a linear layer instead of the attention-classifier framework. We trained each model on the preprocessed dataset for 1,000 epochs using the Adam optimizer with a learning rate of 0.001. We held out sets s01s02, s03s04, s05s02, s06s04 in the original dataset as our test set.

**Attack Setting.** We used the same step size of $\alpha = 0.03$ and ran our AIA attack for $M = 400$ iterations in all experiments. In addition, we used the Adam optimizer with a learning rate of 1e-3 to minimize the adversarial loss function $\mathcal{L}_{adv}$. The scaling factor $\lambda$ for the temporal loss term $\mathcal{L}_{temporal}$ was set to 0.1. The tolerance factor $\kappa$ was selected for each target reaction based on our previous informal user survey in Appendix B.1.

### 4.2. Effectiveness of our AIA Attack

In this experiment, we examine the effectiveness of our AIA attack under the white-box setting with different values of maximum perturbation $\epsilon$ allowed. Successful attacks need to satisfy two conditions: 1) the adversarial output sequences need to be recognizable as the target reaction (related to $\kappa$), and 2) the adversarial input sequences need to be visually

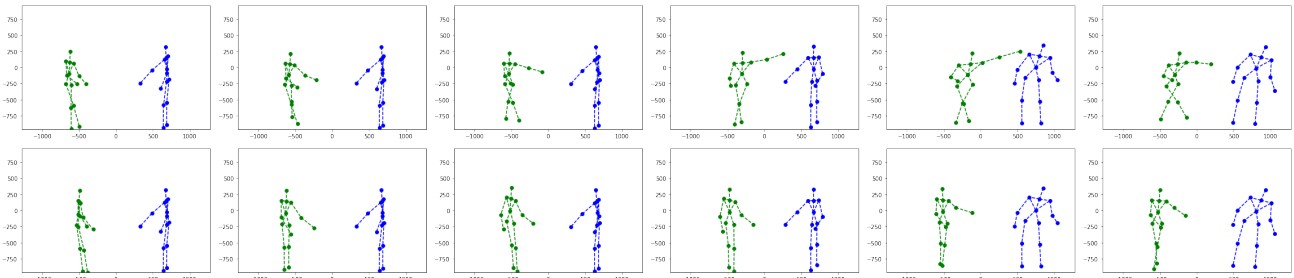

*Figure 1.* Side-by-side comparison of Case Study 'punching' to 'handshaking'. Top-Bottom: original prediction, adversarial prediction. Blue character: input, green character: output.

similar enough compared to the natural input sequences such that it can circumvent security detection (related to $\epsilon$). Hence, the smaller the $\epsilon$ the attack can work under, the more effective the attack is.

To control the overall change to the input sequence, we perturbed only the depth dimension for each joint. This makes it much easier to visualize perturbations. On a side note, this is a stricter optimization problem with constraints compared to the original proposed problem. Thus, the outcome of this experiment is applicable to the original problem as well.

### 4.2.1. ADVERSARIAL TARGETS.

We created 8 sets of target reactions, corresponding to all 8 interactions in the SBU Kinect Interaction Dataset. The objective of each set of targets is to change the output reactions of all test data into one specific target reaction. We then perform targeted adversarial attacks based on these objectives over a range of $\epsilon$ values.

We consider an attack to be successful if the sum term in Equation 1 computed on the test datum is less than the human-determined $\kappa$ based on the sample sets. Otherwise we consider the attack to have failed. The average attack success rates over all 8 target sets under various $\epsilon$ are reported for both models in Figure 2. We used the $\kappa$ sampled from human judges to evaluate attack success rates for objectives 1 to 5. We used the average $\kappa$ over 5 objective sets to evaluate the remaining 3 attack objectives.

### 4.2.2. RESULTS.

On average, with a perturbation factor $\epsilon$ of 0.225 to 0.3, our AIA attack is able to alter almost all output sequences of the DeepGRU model into any target sequence. In contrast, a larger $\epsilon$ of 0.375 to 0.45 is necessary for AIA to achieve a similar level of performance on the TCN model. In general, the TCN model is more robust to our attack than the DeepGRU model. However, under this white-box setting, we were able to achieve a 100% attack success rate

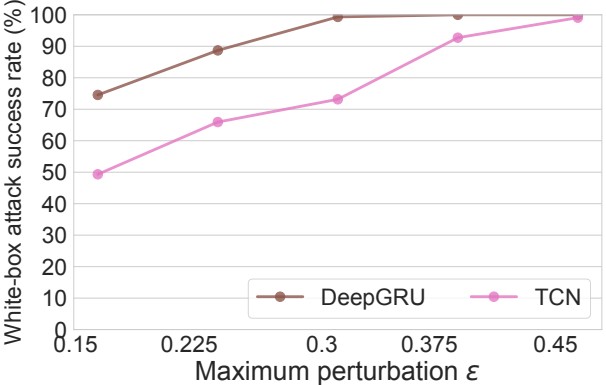

*Figure 2.* Average white-box attack success rate of our AIA attack on TCN and DeepGRU.

on almost all target sets for both models. Overall, our AIA algorithm is able to accomplish most attack objectives with small perturbations of 2% to 5% to natural input sequences.

## 5. Conclusion

In this paper, we presented a framework for attacking skeleton-based interaction prediction models. We proposed the first targeted sequential regression attack that is capable of altering the entire output sequence completely - Adversarial Interaction Attack (AIA). On top of that, we also defined an evaluation metric that can be adopted to evaluate the performance of adversarial attacks on sequential regression problems. We demonstrated on variants of two previous state-of-art action recognition models, TCN and DeepGRU, that our AIA attack is very effective. Additionally, we showed that our AIA attacks are highly transferable if referenced from proper models. We also discussed through a case study, how AIA might impact interactions between human and AI in real scenarios. Future work will look into the extension of AIA to other spatio-temporal regression settings. We hope this serves to motivate careful consideration about how to effectively incorporate AI based agents into human daily life.

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

# A. Extra Case Studies

## A.1. Extra Case Study 1: 'handshaking' to 'punching'

Figure 3 illustrates a successful AIA attack that fools the model to predict a 'punching' action for the reactor (the green character) as a response to the adversarially perturbed 'handshaking' action of the actor (the blue character). Note that the perturbation only slightly changed the actor's action. This reveals an important safety risk that needs to be carefully addressed before machine learning based AI agents can be widely used in human daily life. Suppose that we are at an AI interactive exhibition, a participant would like to shake hands with an AI robot agent. He gradually extends his hand, sending out an interaction request to the AI agent and is expecting the AI agent to respond to his handshaking invitation by shaking hand with him. However, instead of reaching its hands out gently, the AI agent decided to punch the participant in the face because the participant's body does not stay straight. It would be extremely hazardous if the human character unintentionally wiggled his body in a pattern similar to the adversarial perturbation introduced in this case study. While the actual chance of this happening is extremely low due to the high complexity of data in both the spatial and the temporal dimensions, this threat might nevertheless happen if AI workers become widely deployed worldwide. In this case, the human is a victim by inadvertently performing an adversarial attack (wiggling their body).

## A.2. Extra Case Study 2: 'approaching' to 'remaining'

Extra Case Study 2 demonstrated in Figure 4 examines the case of how a cheater might be able to bypass an AI agent's detection. Whilst automatic ticket checkers have been widely adopted, manual ticket checking is still required for numerous situations. For instance, public transportation companies may want to check whether a passenger has paid for the upgrade fee if he or she is in a first class seat. Now suppose that a public transportation company decides to hire AI agents to do the ticket checking job. The public transportation company will lose a huge amount of income if passengers know how to stop the ticket checkers from 'approaching' as in Figure 4, or even change their 'approaching' response to 'departing'.

# B. Empirical Understanding of AIA

## B.1. Tolerance Factor $\kappa$

The objective of AIA attack is defined with respect to a tolerance factor $\kappa$ (see Equation 1, Equation 3 and Equation 4), which is a flexible metric that distinguishes whether the output sequence is close to the targeted adversarial reaction. Because there are many factors involved, such as the char-

acter's height, handedness, and the direction the character is facing, conventional distance metrics such as $L_1$ and $L_2$ norms are not suitable to define precisely what the pattern of a specific action should look like. Therefore, we determine the value of $\kappa$ based on human perception via an informal user survey.

In order to obtain appropriate values for $\kappa$ to evaluate whether an attack is successful, we randomly sampled 5 out of 8 sets of attack objectives and presented them to 82 human judges, including computer science faculties and students. Each objective set is composed of an action-reaction pair and contains output sequences generated from 6 different values of $\epsilon$ (from left to right in ascending order). For each sample set, we asked the human judges to choose the leftmost sequence they believe is performing the target reaction. Sampled objectives and the responses from the 82 human judges are recorded in Table 1.

Based on the responses from the 82 human judges, we computed the tolerance factor $\kappa$ in the optimization problem defined in Equation 1 based on the average of

$$\sum_{t \in T} \|f(\mathbf{x}'_1, \cdots, \mathbf{x}'_t) - \mathbf{y}'_t\|_2 \qquad (7)$$

over the 5 sample objective sets. The calculation of Equation 7 for each objective set is based on the minimum $\epsilon$ polled from the 82 human judges, and the corresponding value of $\kappa$ is then selected as the optimal value (boldfaced in Table 1).

Note that, $\kappa$ serves as a geometrical boundary between the natural and the adversarial *outputs*, whereas $\epsilon$ is a maximum perturbation constraint that we don't want the *input* perturbation to go beyond.

## B.2. Effect of the Temporal Constraint

Here, we study the effect of the temporal constraint $\mathcal{L}_{temporal}$ defined in Equation 5 on the naturalness of the generated adversarial input action sequence. Specifically, we investigate how the input skeleton sequence changes in the depth axis as that is the only perturbed dimension throughout our experiments. Our hypothesis is that this additional factor will enable our AIA attack to find adversarial input sequences that change more smoothly with respect to time.

We demonstrate visually a comparison between adversarial sequences generated with and without the temporal constraint in Figure 5. The top sequence is an adversarial input sequence generated with the $\mathcal{L}_{temporal}$ term removed, whereas the bottom sequence is an adversarial input sequence generated with $\lambda = 0.1$ scaling factor applied to the $\mathcal{L}_{temporal}$ term. In comparison to the previous experiment, we plot the skeletons from the depth-y point of view as we

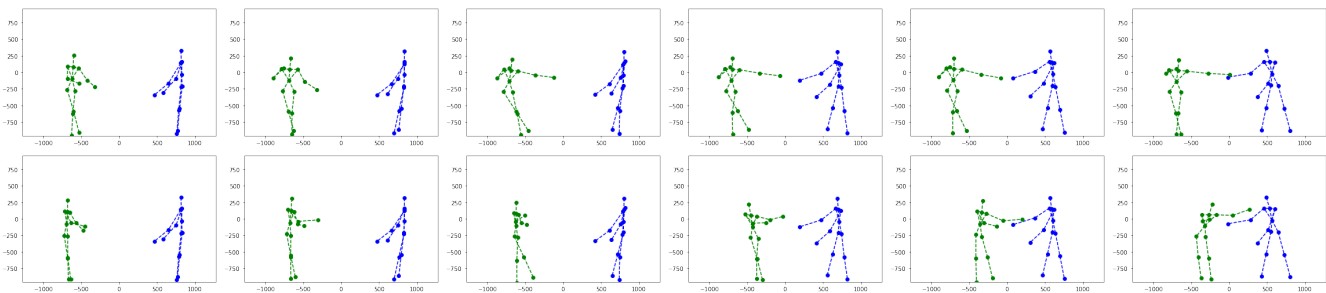

*Figure 3.* Side-by-side comparison of Extra Case Study 1 'handshaking' to 'punching'. Top-Bottom: original prediction, adversarial prediction. Blue character: input, green character: output.

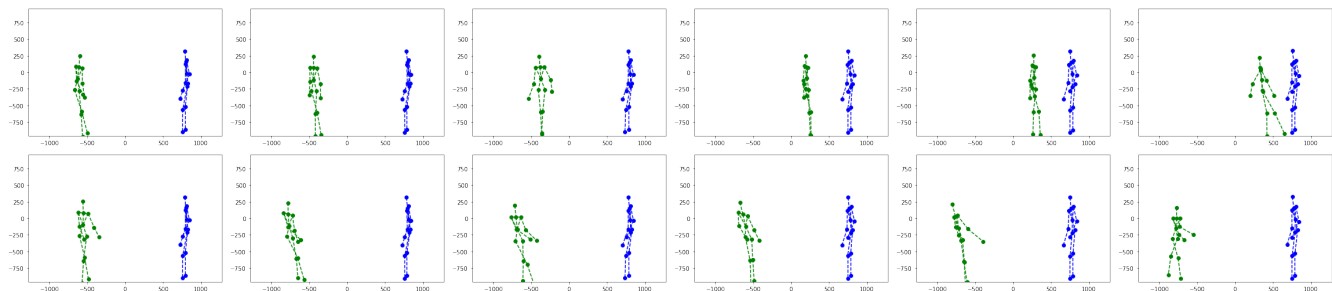

*Figure 4.* Side-by-side comparison of Extra Case Study 2 'approaching' to 'remaining'. Top-Bottom: original prediction, adversarial prediction. Blue character: input, green character: output.

*Table 1.* Responses from the 82 human judges. The optimal $\kappa$ for each attack objective is highlighted in **bold**.

| $\epsilon =$ | 0.075 | 0.15 | 0.225 | 0.3 | 0.375 | 0.45 |
|---|---|---|---|---|---|---|
| Handshaking | 1 ($\kappa = 90.9$) | 4 ($\kappa = 84.28$) | **44 ($\kappa = 79.52$)** | 3 ($\kappa = 74.49$) | 12 ($\kappa = 45.04$) | 14 ($\kappa = 35.03$) |
| Punching | **58 ($\kappa = 52.04$)** | 13 ($\kappa = 47.63$) | 6 ($\kappa = 43.97$) | 3 ($\kappa = 41.76$) | 0 ($\kappa = 39.14$) | 2 ($\kappa = 34.91$) |
| Kicking | 3 ($\kappa = 100.61$) | **71 ($\kappa = 93.17$)** | 7 ($\kappa = 86.57$) | 1 ($\kappa = 80.68$) | 0 ($\kappa = 47.47$) | 0 ($\kappa = 35.36$) |
| Departing | 0 ($\kappa = 85.03$) | 7 ($\kappa = 76.78$) | **26 ($\kappa = 71.77$)** | 12 ($\kappa = 67.58$) | 1 ($\kappa = 41.78$) | 10 ($\kappa = 32.70$) |
| Pushing | 6 ($\kappa = 28.66$) | 3 ($\kappa = 26.55$) | 2 ($\kappa = 25.16$) | 14 ($\kappa = 23.98$) | **49 ($\kappa = 22.77$)** | 5 ($\kappa = 21.31$) |

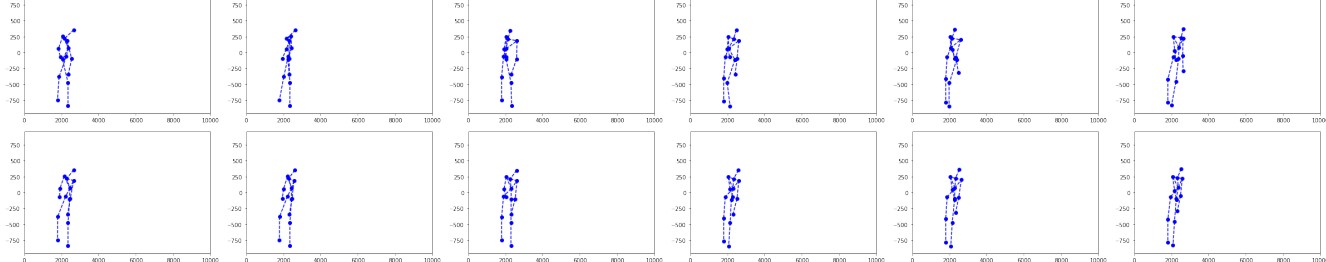

*Figure 5.* Adversarial input action sequences generated by our AIA attack with (bottom row, and $\lambda = 0.1$) or without (top row) the temporal constraint $\mathcal{L}_{temporal}$.

are more interested in visualizing the perturbation.

As shown in Figure 5, it is observable that in general, the top sequence has more abrupt changes in body position between each time step. This almost never happens in the bottom sequence. More specifically, in the bottom sequence, when a larger change to the body posture is necessary, the change is always preceded by smaller changes in the same direction. In contrast, in the top sequence, any large changes can take place in just one time step. This type of aggressive change should be avoided as much as possible, as it could make the

attack more easily detectable.

## C. Extra Experiment on Black-Box Transferability

In addition to white-box effectiveness, we examine how transferable our attack is. An adversarial example generated based on one model is said to be transferable if it can also fool other independently trained models. In this experiment, we examine robustness of the TCN model and the DeepGRU model towards adversarial examples generated based on each other.

### C.1. Black-box Setting.

We employed the same metric established in Section 5.1 to determine an attack to be successful or not. To evaluate how strong our attack is under the black-box setting, we reused the adversarial input sequences in the previous experiment. We feed all adversarial sequences generated based on one model into another and inspect their effectiveness when used to attack unseen model. In other words, we use adversarial sequences generated based on the DeepGRU model into the TCN model and vice versa. The average black-box attack success rates over a range of $\epsilon$ are reported for both models in Figure 6.

### C.2. Results.

Surprisingly, adversarial examples generated from the TCN model are remarkably strong. With an $\epsilon$ value of 0.375 to 0.45, adversarial actions generated from the TCN model successfully fooled the DeepGRU model more than 80% of the time for almost all attack objectives. Along with the results in Section 4.2, this substantiates that our AIA attack has high transferability in addition to being effective.

We also observed that adversarial actions generated from the DeepGRU model are rather weak on the TCN model under the black box setting. It is only able to achieve an average success rate of 30% irrespective to the maximum perturbation $\epsilon$ permitted. The TCN model is more robust than DeepGRU in the white-box setting. We suspect that this is because the convolutional layers used in TCN are more robust than the gated recurrent units of DeepGRU. Specifically, in order to fool the TCN model, the attack needs to take into account the high level feature maps between the convolutional layers. However, adversarial examples generated from the DeepGRU model might not be able to fool the convolutional layers of TCN because these high level features were not taken into consideration in the first place. Note that, while being relatively more robust, TCN also leads to more transferable attacks. We leave further inspection to this disparity as a future work.

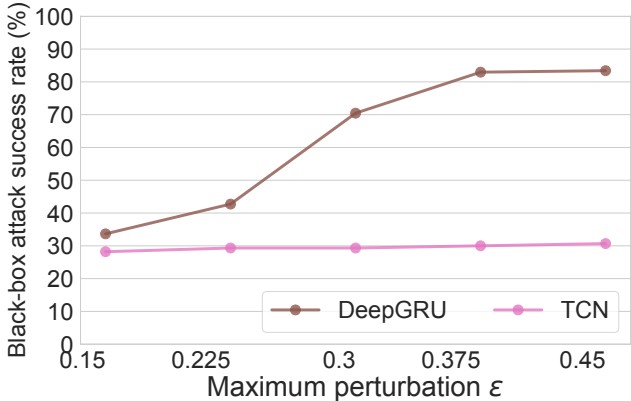

*Figure 6.* Average black-box attack success rate of our AIA attack on TCN and DeepGRU.