# OpenReview forum: "Adversarial Interaction Attacks: Fooling AI to Misinterpret Human Intentions"
_ICML.cc/2021/Workshop/AML — ICML 2021 Workshop AML Poster_

### Official Review · Reviewer_uobt · 2021-06-19

**Rating:** Accept
**Confidence:** 4

**Review:**

This paper proposes an adversarial interaction attack method to fool skeleton-based action interaction models. The method involves a spatial loss and a temporal loss to improve the similarity and naturalness of the generated adversarial examples. The experiments on several case studies show the effectiveness of the proposed method.

This paper is well-written. The proposed method is effective. One thing may be interesting is that can the proposed attack be realized in the real-world.

---

### Decision · Program_Chairs · 2021-06-21

**Decision:**

Accept (Poster)

**Comment:**

A good work for adversarial attacks on skeleton-based action interaction models. The paper is well-written.